# Spatio-Temporal Action Detection in Untrimmed Videos by Using Multimodal Features and Region Proposals

**DOI:** 10.3390/s19051085

**Published:** 2019-03-03

**Authors:** Yeongtaek Song, Incheol Kim

**Affiliations:** Department of Computer Science, Kyonggi University, 154-42 Gwanggyosan-ro Yeongtong-gu, Suwon-si 16227, Korea; dudtroc92@kyonggi.ac.kr

**Keywords:** video action detection, region proposal, spatio-temporal action detection, recurrent neural network

## Abstract

This paper proposes a novel deep neural network model for solving the spatio-temporal-action-detection problem, by localizing all multiple-action regions and classifying the corresponding actions in an untrimmed video. The proposed model uses a spatio-temporal region proposal method to effectively detect multiple-action regions. First, in the temporal region proposal, anchor boxes were generated by targeting regions expected to potentially contain actions. Unlike the conventional temporal region proposal methods, the proposed method uses a complementary two-stage method to effectively detect the temporal regions of the respective actions occurring asynchronously. In addition, to detect a principal agent performing an action among the people appearing in a video, the spatial region proposal process was used. Further, coarse-level features contain comprehensive information of the whole video and have been frequently used in conventional action-detection studies. However, they cannot provide detailed information of each person performing an action in a video. In order to overcome the limitation of coarse-level features, the proposed model additionally learns fine-level features from the proposed action tubes in the video. Various experiments conducted using the LIRIS-HARL and UCF-10 datasets confirm the high performance and effectiveness of the proposed deep neural network model.

## 1. Introduction

Recently, owing to the wide use of video-sharing sites, the production and consumption of video data have been rapidly increasing. Accordingly, studies are actively conducted to automatically analyze video contents and detect human actions contained in them. The video-based human-action-detection technology refers to (1) not only the estimation/classification of human actions contained in a video (action classification, AC) (2) but also the detection of temporal regions of a video containing the respective actions (temporal localization). Recently, as shown in Figure 1, (3) owing to the detection of a spatial region showing humans performing an action in a video (spatial localization), spatio-temporal human-action-detection technology, which enables the detection of actions of each person included in a video, has been gaining interest [1,2,3]. Such a human-action-detection technology can be used in various applications in both indoor and outdoor environments, e.g., video-based surveillance, security and search systems, VOD (Video On Demand) data management systems, artificial intelligence social robotic systems, and elderly care service systems.

In the early days, video-action-detection research dealt with the problem of classifying only one action of a single person contained in a short video (video classification) [4,5,6]. However, at present, when existing technologies are used for the video AC problem, high performances are demonstrated for the video sets distributed online. However, finding videos containing an action of one person only from a vast number of videos used in the real world not easy. Therefore, the spatio-temporal video-action-detection problem should be solved to find all multiple action regions starting and ending asynchronously in an untrimmed video, and the action type in each region should be recognized.

To solve spatio-temporal action-detection problem, various methods have been developed in recent years [1,2,3]. An action occurs across many frames/clips rather than just one frame/clip. Therefore, the conventional method of classifying an action independently in each frame/clip unit is not suitable for the detection of a continuous action. To supplement the above-mentioned problem, a method of using context information between video images was proposed. Particularly, Escorcia et al. [7] and Ma et al. [8] proposed a method to effectively obtain context information by using a recurrent neural network (RNN) [9], which is a type of deep neural network. However, the methods of using context information are still limited in finding a spatio-temporal boundary region of each action. Recently, two-dimensional (2D) object-detection technologies, such as Faster RCNN [10], and Single-Shot MultiBox Detector (SSD) [11] were used to detect an object in a particular region of an anchor box in a video image, thereby proposing an efficient framework that classifies objects contained in a corresponding region. By applying these region-proposal techniques to a three-dimensional (3D) video including time, the model proposed in this paper presents methods for detecting a spatio-temporal region contained in a video. Unlike a 2D-object-detection model that detects an object from a single image, the proposed video action-detection-model deals with the difficult problem of understanding a 3D continuous-state space to detect actions.

In addition, as this method detects an action by using given video-image features, emphasis has been placed on the effective feature learning methods using videos. Recently, by using the convolutional neural network (CNN), which is a type of deep neural network, coarse-level feature information, which contains comprehensive information across the whole video, has been mainly used [12,13]. Coarse-level features usually require the learning of optical-flow information, which changes between images [14], or spatial information, such as objects and background appearing in images [15]. Lately, semantic features are also often used, whereby words are used to clearly express action objects appearing in a video [16]. Such semantic features that can express high-level concepts can contribute to the performance improvement of action detection. However, although coarse-level features are learned from a whole video, they are still unable to provide information directly related with each person performing an action. To overcome the limitation of coarse-level features, fine-level features focused on specific regions of interest (ROI) of the given video have been additionally used for action detection. Therefore, after several regions of interest (ROI) are first proposed, and fine-level features are then extracted from the proposed regions. In recent studies, a variety of fine-level features have been suggested for their own unique purposes. The study in [17] tries to learn fine-level features representing high-order inter-object interactions in a video, and then to make use of these fine-level features for action recognition and video captioning. 

The proposed model of this paper learns both coarse-level features and fine-level features from a given video. While coarse-level features encode context information about each video frame, fine-level features encode detail information for the proposed action tubes. Our model uses human pose information extracted the proposed action tube as fine-level features. Particularly, when the pose information is accumulated by the clip size, the trajectory information can be obtained. Trajectory information is effective in determining the primitive action. To overcome the fragmentation problem, that the features learned from adjacent video clips are independent of each other, the proposed model uses an RNN. Furthermore, among the conventional RNN models, the bidirectional long short-term memory (BLSTM), which is an extended version of LSTM, is used to learn the feature information sharing context of adjacent video clips. For more effective temporal region detection, our model uses a complementary two-stage temporal-region proposal method. By using the two-stage temporal region proposal method, our model improves both action detection accuracy and detection time.

Recently, with the increase in the interest in video-based action detection, a large number of various open datasets have appeared online. In the early days of action detection research, UCF-101, which is an action dataset related to daily lives, was mainly used for performance measurement. However, in recent years, datasets belonging to various categories, such as sport and primitive action, have become publicly available, e.g., THUMOS, ActivityNet, Charades, AVA, and LIRIS-HARL. In this study, the model performance is evaluated using both UCF-101 and LIRIS-HARL data sets. Each video of the data sets contains multiple spatio-temporal actions.

In summary, this study makes the following contributions to relevant literature.
In order to detect effectively multiple concurrent spatio-temporal actions in an untrimmed video, the proposed model learns both coarse-level features and fine-level features in a complimentary manner. Based on coarse-level features encoding context information about each entire video frame, the model proposes multiple spatio-temporal action regions, which result in multiple action cube proposals. And then, our model extracts human pose information as fine-level features from the proposed action cubes, which encode rich information for action detection and classification.To overcome the fragmentation problem, the proposed model uses the bidirectional long short-term memory (BLSTM), which is an extended version of the LSTM, to learn the feature information sharing context of adjacent video clips.Finally, our model adopts a two-stage temporal region proposal method. The two-stage temporal region proposal method improves both action accuracy and detection time.

The remainder of this paper is organized is as follows. Section 2 examines the related literature briefly. Section 3 describes the overall structure of the proposed model, and Section 4 introduces the datasets used for training and evaluating the model and describes the method of implementing the model. Section 5 discusses the results of experiments conducted for evaluating the model performance. Lastly, Section 6 presents the conclusion and future scope.

## 2. Related Work

### 2.1. Object Detection in Images

In this study, an object-detection technique was used for effective video-based action detection. In the earlier studies on image-based object detection areas, objects in an image are classified using a simple classification method (object classification). However, when several objects appear simultaneously in an image, classifying objects through simple classification is not possible and the location information of objects cannot be identified. To solve these problems, recent object-detection methods propose candidate regions of fixed size and classify an object in that candidate region. Typical models of object detection include Faster R-CNN [10], SSD [11], and YOLO [18]. These models all propose candidate regions, and by assigning scores to the regions containing objects from among the proposed regions, select a candidate region with the highest score as a representative proposal region. By applying the proposed method to 3D video regions including time and by proposing regions for a certain time length, these models can classify if a proposed region contains an action. Furthermore, to detect an action with respect to an individual appearing in a conventional 2D image, the conventional object detection model, SSD, was used. Thereafter, by using the detected spatial location information, fine-level feature information was learned.

### 2.2. Human Pose Estimation

To utilize fine-level information of each person performing an action in a video, the proposed spatio-temporal action-detection model uses human pose information, i.e., fine-level feature information. The body joint information of each person, which changes according to time, was detected in the video, and then this information was used to determine the motions of each person. The video-based human-pose-estimation method consists of the following processes: the detection of a human appearing in an input video (human detection), estimation of joint location of the detected human (human joint localization), and finally estimation of human poses by using the joint information (human pose estimation). Previous studies mainly focused on estimating 2D-based joint location by using an RGB image as an input [19,20,21]. However, depth information cannot be utilized in an RGB image, and subsequently solving an occlusion problem, i.e., the problem in which the target object is occluded by another object, is difficult. Furthermore, Tome et al. [22] proposed a method to estimate the 3D pose of a human appearing in a video; however, their method was limited to the conventional feature information that could be obtained from RGB images. To supplement this problem, methods were proposed to solve the occlusion problem by using depth-information-added RGB images [23,24] or by using two or more cameras [25]. The recent methods of human pose detection can be mainly classified into the top–down and bottom–up approaches. First, the top–down approach is a conventional method of detecting human poses; it detects a pose by extracting information from high- to low-level evidences [20,22,26,27,28]. Usually, a human pose is detected in the following sequence: first, a human appearing in a video is detected, and then the human joint-location information is estimated. This has the advantage that information can be systematically collected through the sequential approach. However, in recent years, the bottom–up approach has accomplished higher performance output, whereby a human pose is estimated by calculating the correlation between body joints [19,29,30]. In the case of the bottom–up approach, unnecessary computations were excluded during the detection of humans appearing in a video. Accordingly, if the number of persons appearing in a video increases, the detection speed increases slowly compared with that in the top–down approach. 

### 2.3. Action Detection in Videos

Previous studies of video-based action detection were conducted to classify the actions contained in short trimmed videos, which were already divided based on actions [12]. Usually, a video was divided in clip units of a certain size and an action contained in each clip was determined. However, untrimmed videos used in real life contain actions with different starting and ending times. Therefore, to detect actions in an untrimmed video, it is necessary to perform not only the classification of multiple actions contained in the video but also the action temporal localization (i.e., finding a temporal region in which each action is contained). When the conventional action-classification method of clip units is used, fragmentation tends to occur with respect to detection of temporal regions of actions [31]. To overcome this problem, a method of reflecting context information between adjacent clips was proposed; this method has usually been used with RNNs [32,33,34]. As another method of reflecting context information, a smoothing technique has been also used often; this technique uses action score values given to each video/clip and obtains an average score of the adjacent videos/clips [30]. However, when context information is used, finding the respective spatio-temporal boundaries of multiple actions in an untrimmed video is difficult.

To solve this problem, similar to SSD and faster R-CNN techniques for 2D image-based object detection, efficient frameworks have been also proposed for detection within a 3D video-based action area, whereby action candidate boxes (anchor boxes) of various lengths are proposed from a video (temporal region proposal, TRP); based on these boxes action-classification is performed [1,35,36]. This method can detect a temporal boundary of each action effectively as it classifies fine-level actions bounded by the size of the anchor box. However, as the box size becomes increasingly diverse, the number of boxes to be generated increases considerably. Furthermore, when generating an anchor box, it could overlap with another anchor box generated earlier, and this could lead to the generation of an increasing number of boxes. When the number of anchor boxes increases, a region containing an action can be investigated in more detail but if the number is considerably high, action detection could be delayed. For the temporal-region-proposal process, this paper proposes an efficient framework that filters regions not containing any action according to the clip units before the generation of anchor boxes of a fixed length.

Several works attempt to simultaneously localize not only temporal but also spatial regions of action. In a typical spatio-temporal action detection research, human spatial regions appearing in each image of a video are detected and the type of actions included in the proposed region are classified [37]. Recent work [1,2] detects and classifies the spatio-temporal area of an action by calculating associations between subsequent images.

On the other hand, learning effective features from a video is also an important issue. Most of the action detection researches used coarse-level features that contain comprehensive information of an image using CNN. Recently, however, studies have researched to detect human action based on pose information of people detected in the video. The 2D joint position of human is detected on a frame-by-frame, and the trajectory information of the joint, which changes with time, can be obtained by accumulating each detected joint information by a clip length [27]. However, 2D pose information is difficult to consider action information at various points of view. In this paper, we use 3D pose information which can consider more diversified information than 2D pose information. 

To investigate this further, various experiments were performed using UCF-101 and LIRIS-HARL, which are open benchmark datasets available online. Through these, the performance and effectiveness of the proposed deep neural network model were analyzed.

## 3. Action Detection Model

This paper proposes a deep-neural-network-based video-action-detection model, the structure of which is shown in Figure 2. In terms of functions, the model mainly consists of feature learning, region proposal, and action-classification processes. First, the proposed model has separate processes for learning coarse- and fine-level features. In the process of coarse-level feature learning (CFL), visual and semantic features, which are often used in conventional action-detection methods, are learned. A coarse-level feature contains comprehensive information of an image, and in the proposed model, such features were learned in clip units. 

Later, through the feature-sharing process, in which information is exchanged with surrounding clips, multimodal features were generated. By commonly using the learned multimodal features in the subsequent TRP and AC processes, the feature-learning time can be reduced and simultaneously consistent results can be obtained. In the fine-level feature learning (FFL) process, first, by going through the spatial region proposal (SRP) process, the region where each person appears in an image is detected. Next, by using these detected regions as inputs, the pose information of each person is learned in the FFL process. The TRP process of the proposed model consists of two complementary stages. First, the regions predicted to potentially include actions are filtered, and then anchor boxes of various sizes are generated targeting the remaining regions so that actions of various lengths can be detected. Finally, in the AC process of the proposed model, the multimodal and pose features targeting the regions of anchor boxes proposed earlier are used to calculate the AC scores of the detected persons. Among the scores of the respective actions calculated in clip units, an action with the highest average score is classified as an action representing the corresponding anchor box. 

### 3.1. Coarse-Level Feature Learning

The proposed model learns two types of visual features from a given input video, as shown in Figure 3. First, a 3D CNN, C3D was used to learn the optical-flow information that can help recognize a dynamic action itself in continuous frames. In addition, a 2D CNN, VGG-16 was used to learn the visual features that can be used for detection of objects related with an action appearing in a video. By using the respective networks, two feature vectors with size 4096 and 1000 were learned. Furthermore, the proposed model additionally learned the semantic features that can represent the video contents in high-level concepts. Semantic features consist of the weights of respective words calculated according to the action and object types appearing in the image frames. Compared with visual features containing comprehensive information of conventional images, sematic features can provide clear information related with an action. The proposed model was trained using 1024 dynamic semantic features and 1024 static semantic features. Next, to solve the fragmentation problem, which occurs as the feature vectors learned in clip units are disconnected from the information of adjacent clips, BLSTM, a type of bidirectional RNN, was used in this study, as shown in Figure 3. In the case of RNN, GRU, and LSTM, which have been conventionally used, only the past information is used as information is delivered unidirectionally. However, to detect actions contained in a video, i.e., to determine an action of a current clip, both future information as well as past information are needed. For example, suppose videos of a pole vault and long jump are compared.

Then, when only the running scenes in the initial part of a video are watched, determining what these actions are, based on the past and present information, will be difficult. However, if future information is also used to determine the current action, this problem can be solved. In this paper, BLSTM was used to learn the multimodal features that share context information of adjacent clips. Multimodal feature vectors have a size value of 1024, and they are used in the subsequent TRP and AC processes. Simultaneously, by understanding the before and after contexts, the feature-learning time can be reduced and consistent results can be obtained.

### 3.2. Temporal Region Proposal

For the spatio-temporal action detection in a video, it is necessary to not only classify the actions contained in the video but also effectively detect the temporal region boundary of each action contained in the video. Various action-detection methods have been proposed to detect actions of variable lengths. By applying the 2D-object detection methods mentioned in Section 2.3 to the 3D video area, methods that generate anchor boxes of various lengths have been provided by targeting temporal regions. These methods can easily detect actions of various lengths. However, as anchor boxes are generated across whole regions of a video, a large amount of calculation is required in the AC process. Therefore, to reduce the amount of calculations and detect actions with high accuracy, this paper proposes a complementary two-stage TRP method, as illustrated in Figure 4.

As shown in the figure, the first stage uses the multimodal features, which were learned previously from the video images, as inputs of the no-action-clip filter. As the multimodal features contain the information of adjacent clips, this method has the advantage of preventing fragmentation of a single action over distributed multiple clips. The structure of the no-action-clip filter consists of a fully connected (FC) layer and a Softmax classification layer, and score values are assigned to each clip based on whether it contains an action. The output score values are compared with a threshold of TRP scores (tTRPS), and when the clip score is high, the clip is determined to potentially include an action. As shown in the middle part of Figure 4, through the first stage of TRP, clip regions are classified into those containing actions (yellow boxes) and not containing actions (white boxes). In the first stage, as anchor boxes are generated to target the filtered regions, the anchor-box classifications and calculations can be reduced. Furthermore, when the AC is performed, as unnecessary context information is excluded for the regions in which actions do not occur, actions can be detected effectively. However, in the first-stage process alone, the detection of actions of various lengths in the determined temporal regions is difficult. To supplement this problem, anchor boxes of various sizes were generated in the second stage by targeting the regions containing potential actions, which were obtained through the filter. As anchor boxes of 4, 8, and 16 clips were generated, detection of actions of various lengths is possible. Similarly, in the TRP-process structure of the proposed model, the respective stages complement each other.

### 3.3. Spatial Region Proposal

The conventional action-detection methods focused on classifying actions contained in videos and detecting the time when the actions occurred. However, the conventional studies have a crucial problem: in the conventional studies because only coarse-level feature information is mainly learned, which is obtained across a whole video, individual information of persons appearing in the video cannot be used. Therefore, in this study, the SRP process was given importance. In this process, human regions are detected in a video by using the conventional object-detection method to identify the location of the principal agent performing an action in the video. Next, the detected spatial information can help learn pose features (fine-level feature information introduced in Section 3.4) to obtain information directly related with human actions.

To detect multiple persons appearing in a video, the SRP process of proposed model uses an SSD (Single Shot Multibox Detector), which is an object detection model, as shown in the lower left side of Figure 2. The SSD detects the majority of people appearing in the video with high accuracy. However, when a person turns around or assumes an abnormal pose, the detection fails. If the number of video regions in which human detection is missed increases, fragmentation occurs. If the occurrence of fragmentation increases, the accuracy of human detection decreases. To supplement the missing detection information, the proposed model tracks human region information. If the region of intersection (RoI) between the regions before and after the region where human detection was missed has a value higher than the threshold value, the region where the detection of a person was missed is tracked. Through the SRP process, the proposed model obtains the location information of the person performing an action in each video, and learns the pose information (fine-level feature information). Section 3.4 introduces the method of learning pose information.

### 3.4. Fine-Level Feature Learning

In the conventional action-detection methods, coarse-level features learned from a whole video were mainly used to classify a human action. However, suppose two students on a playground are performing different actions while wearing same clothes. To classify their respective actions, features will be learned from an input video. In the case of conventional coarse-level features obtained across the whole video, the spatial feature information, such as background or nearby objects in the video, or visual feature information, such as visual flow information between images, can be used. However, if only these features are used, the classification of the actions of two students moving in a same background while wearing same clothes will be difficult. The coarse-level feature information learned from a whole video has a limitation in that the individual feature information of each person, independent of each other and directly related with the respective action, cannot be provided. To provide the feature information directly related with the actions of the persons appearing in the video, this study used human pose information. As pose information provides trajectory information of joints changing by time, it is effective in detecting high-level actions as shown in Figure 5.

To learn the human pose feature information for action detection, the proposed model uses the pose feature network proposed in [21], which can be used to estimate the 2D and 3D pose information from an RGB input video. In general, there are mainly two problems with respect to estimating human pose information from a video. First, when estimating a 2D pose, obstacles exist owing to changes in visual appearance, such as the recognized human’s body size, color of clothes, surrounding lighting, and viewpoint of camera. Second, the number of cases for converting estimated 2D pose information to 3D pose information can be infinite instead of only one. To solve these problems, the model of [21] uses a structure of a convolutional pose machine (CPM), which has a multistage structure. CPM extracts features from an input video in every stage, and outputs an estimated pose map (belief map) for each human joint. Furthermore, the biggest characteristic of CPM is that it has a structure for improving the accuracy of the belief map of each joint continuously by using the outputted belief map of the previous stage as the input of the current stage. Furthermore, the model of Tome et al. [21] uses a Gaussian 3D pose model to estimate the 3D pose information from the 2D pose information obtained from an RGB video. In addition, by comparing the Gaussian model with the built-in 3D pose model, 3D pose information with the closest consistency is estimated from the estimated 2D pose information.

To remove the obstacles during action detection by focusing more on the background or objects, the proposed model learned the human pose feature information. Also, we use 3D pose information which can consider more diversified information than 2D pose information. The model of Tome et al. [21] cuts an RGB image into respective human units through the SRP process and uses them as inputs. For the pose feature information, the location information of each human joint in an image is obtained. The proposed model detected a total of 18 3D joint points including those located at the head, neck, shoulder, elbow, hand, abdomen, hip, knee, and ankle. In the case of visual and semantic features used in the conventional studies, as feature information is learned across the whole image, during action classification, actions are detected by focusing more on the surrounding background or objects rather than the person performing an action. However, in the case of pose feature information, the trajectory information of joints, changing according to time, can be identified by using respective joint-point information that changes with respect to every image. Such trajectory information can be used for the estimation of current human motion, and the motion becomes the most primitive action that composes an action. A motion pattern can provide important information that can be used when classifying two actions occurring in the same place. However, using only pose information is not enough for classification all the actions. For example, in order to classify other actions having the same type of action, surrounding background or object information must be considered together. Therefore, in this paper, the coarse-level and fine-level features are used together to obtain complementary effect.

### 3.5. Action Classification and Localization

As the inputs of the AC network (ACN), the AC process of the proposed model uses the multimodal features learned earlier using the temporal regions of proposed anchor boxes and the pose information changed during a clip (Figure 6). As shown in Figure 6, the ACN consists of FCs and Softmax layers, which are last layers of the proposed model, and classification score $S=\left(a_{0},a_{1},\ldots a_{n}\right)$ is calculated for the action in each clip unit. For each action, the calculated score is compared with the action threshold (δ). When the score is larger than the threshold value, the action is determined to be a human action contained in the corresponding clip, and by obtaining the average of all the scores calculated for each clip region, an action represented in an anchor box is classified, as shown in Equation (1). In the video-action classification process, by using the previously learned multimodal features, the feature-learning time for AC can be reduced. Furthermore, as the multimodal features and the detected human pose features are used simultaneously, the context information of adjacent images and the changing human joint trajectory information can be used simultaneously.
(1)$$a_{anchor\_box\_main\_activity}=\arg\underset{a_{j}}{\max}\frac{1}{T}\overset{T}{\underset{i=0}{\sum }}a_{j}t_{i}$$

Finally, the action detection process makes use of the classification results of the anchor boxes. Overlapping anchor boxes with the same action type are removed. Each remaining anchor box implies the region of one unique action within the video. As a result, our model localizes the spatio-temporal region of each action within a given video and classifies it into one of the pre-defined action types. 

## 4. Implementation

To conduct a performance test of the proposed model, the deep neural network proposed in this paper was implemented based on Keras and Tensorflow, which are Python deep learning libraries, in the Ubuntu 14.04 UTS environment. The training of and experiment on the model were performed in a hardware environment of 4.0 GHz, 4 core, 8 thread CPU, and Geforce GTX TITAN X GPU card.

### 4.1. Datasets

In this paper, we use two datasets, UCF-101 and LIRIS-HARL, to evaluate the action detection performance of the proposed model. These data sets contain multiple spatio-temporal actions in untrimmed videos. 

*UCF-101*: the UCF-101 dataset provided by the University of Central Florida (UCF) is a large dataset of untrimmed videos with 101 sports action classes of different persons. The proposed model was trained and tested using only 24 classes that provide the ground truth action cubes and labels from the UCF-101 dataset. The dataset consists of 2282 and 910 training and testing videos, respectively. 

*LIRIS-HARL*: the LIRIS-HARL dataset provides a set of 167 video data with 10 action classes of different persons. The dataset provides untrimmed videos for each action, and the average length of videos is short, i.e., 2–3 s. The proposed model used the LIRIS-HARL dataset for training and testing.

### 4.2. Model Learning

The multimodal feature model and TRP network proposed in this paper were trained simultaneously. RMSprop was used as the optimization algorithm, and the binary cross entropy was used as the loss function, as shown in Equation (2). After setting the batch size as 50, epoch as 20, and learning rate as $10^{-2}$, the network learning was performed.
(2)$$\begin{array}{cc} L_{Binary\;Cross\;Entropy} & \\ & =-\frac{1}{N}\overset{N}{\underset{i=1}{\sum }}y_{i}\log\left(h_{\theta }\left(x_{i}\right)\right)\\ & +\left(1-y_{i}\right)\log\left(1-h_{\theta }\left(x_{i}\right)\right) \end{array}$$

The SRP network used in this study detected the spatial information of humans appearing in a video by using the SSD, which is a conventional object-detection model. For the pose feature network that detects subsequent pose information of humans, the CPM structure model of Tome et al. [21] was used.

The last ACN, which composes the proposed model, consists of FCs. Here, Adam was used as the optimization algorithm, and categorical cross entropy was used as the loss function, as shown in Equation (3). Network learning was performed after setting the batch size as 50, epoch as 20, and learning rate as $10^{-2}$
(3)$$L_{Categorical\;Cross\;Entropy}=-\frac{1}{N}\overset{N}{\underset{i=1}{\sum }}t_{i}logy_{i}$$

## 5. Experiments

This section presents the investigation on the spatio-temporal action-detection performance for each class of videos. Moreover, the section presents experiments on spatio-temporal action-detection performance using feature combination and the TRP method. A comparative performance experiment was then conducted by considering the latest action-detection models. The action-detection performance of the proposed model was measured using both UCF-101 and LIRIS-HARL datasets, providing the ground truth spatio-temporal regions of individual actions. The spatio-temporal action-detection performance was measured using the metric IOU_ST_ shown in Equation (4) by calculating the 3D overlap between the ground truth action tube ($ST_{G}$) and the estimated action tube ($ST_{E}$):(4)$$IOU_{ST}=\frac{Area\;of\;Overlap}{Area\;of\;Union}=\frac{ST_{G}\cap ST_{E}}{(ST_{G}\cup ST_{E})-ST_{G}\cap ST_{E}}.$$

In the first experiment, the spatio-temporal action-detection performances between different combinations of features were compared to verify the effect of combining the coarse (CFL)- and fine(FFL)-level features. Accordingly, in this experiment, the same region proposal process and the AC process proposed were performed but not the FFL process. Table 1 shows the changes in the action-detection performance according to the change of tIoU. These were examined by increasing the threshold value (tIoU) from 0.1 to 0.5 in 0.1 steps. For the UCF-101 data set, the background is not fixed. Therefore, using CFL that can understand the context rather than action detection using only the FFL feature has got higher action performance. However, for LIRIS-HARL datasets, the background is fixed and different actions occur in the same place. As shown in Table 2, using FFL, which can provide independent information for each person, was better than using only CFL learned from the whole image. As a result, the experimental results of Table 1 and Table 2 show a higher spatio-temporal action-detection performance when the fine-level features were used in addition to course-level features than when only using one features. As shown in Table 1, when the CFL and FFL are used separately, the mAP values are 39.1 and 22.2. However, when CFL and FFL are used together, the mAP value shows higher performance at 50.3. Table 2 also shows that, when CFL and FFL are used separately, the mAP values were 24.2 and 37.0. However, when CFL and FFL are used together, the mAP value shows higher performance at 44.1.

The second experiment was conducted to analyze the effect of the feature sharing process on the action-detection performance. The BaseLine model does not exchange feature information between adjacent clips. The experiment was proceeded by increasing the threshold value (tIoU) from 0.1 to 0.5 in 0.1 steps. As can be seen from the experimental results in Table 3, both the spatio-temporal action-detection performances of LSTM and BLSTM, which can share features between neighboring clips, are higher than that of the BaseLine model. Also, as we expected, the performance of BLSTM utilizing both information of the past and the future clips is higher than that of LSTM using only information of the past ones. 

The third experiment was conducted to analyze the effect of the proposed TRP process on the spatio-temporal action-detection performance. To this end, Table 4 shows the comparison between spatio-temporal action-detection performances (mAP) by increasing the TRP score threshold (tTRPS) of the TRP process from 0 to 0.45 in 0.15 steps. Here, the same processes of the proposed model were performed excluding the TRP process. Particularly, in the case of threshold tTRPS > 0, all the video clips were determined as effective action regions. Therefore, in the case of tTRPS > 0, anchor boxes are generated for all regions, as in the conventional studies. The experimental results of Table 4 show that in the case of threshold tTRPS > 0.30, high detection performance of approximately 50.3% on average was measured. Therefore, when the regions that were expected to contain potential actions were filtered first by using tTRPS, the spatio-temporal action-detection accuracy increased. However, as shown in the case of tTRPS > 0.45, if the tTRPS increases considerably, the performance will decline.

In the fourth experiment of action detection, the effect of TRP process on the spatio-temporal action-detection processing speed was analyzed. To this end, by filtering the regions containing actions in the first stage of the TRP process and the regions not containing them, the obtained video-based action-detection speed was examined. Table 5 shows the results of measuring the processing speed in milliseconds based on 10,000 frames according to tTRPS. In the case of tTRPS > 0, AC was performed for all the video regions, similar to that in the conventional method. The experimental results showed that when tTRPS > 0.45, the AC processing speed was increased by approximately 80 times. Therefore, it is proven that the exclusion of unnecessary regions is effective for the improvement of processing speed.

In the fifth experiment, a confusion matrix table of AC was used to conduct the AC performance test, as shown in Figure 7. In this table, the correlation between action classes can be confirmed for the UCF-101 dataset, and the results are consistent with those in Table 1. When using the action-class videos on the left side as inputs and calculating the score for each action through the AC process, the estimated action scores are shown for the respective action classes at the bottom of Figure 7. The right side of Figure 7 shows the degree of AC in color. Most classes have a classification accuracy that matches the action label. However, in the case of “SoccerJuggling” or “GolfSwing”, the fine-level feature information could not be actively used because the locations where the actions occurred were very similar, and at the same time, human detection was missed frequently in the process of learning the fine-level features. 

Table 6 shows the video-based spatio-temporal action-detection performance by using the UCF-101 dataset for the first experiment. Threshold θ (threshold IoU, tIoU) on the left side of table shows the minimum value for determining the success/failure of action detection by comparing with the calculated IoU value. Table 6 shows the detection performance of each class according to the tIoU, and the performance was measured by increasing the tIoU value from 0.1 to 0.5 in 0.1 steps. The comparison of Figure 7 confirms that the AC performance of a frame unit directly influenced the video-based action-detection performance. Particularly, in the case of “SoccerJuggling” action, which had a low AC accuracy, the accuracy of the detection performance was low as well. In contrast, in the cases of “BasketballDunk” and “IceDancing” action videos, high action-detection performances were observed. Therefore, the increasing of the AC accuracy of each image composed in a video can increase the action detection accuracy.

In the sixth experiment of action detection, the proposed model was compared with other latest models. The experiment was performed by increasing the tIoU value from 0.1 to 0.5 in 0.1 steps. In the case of other models, some test performances were omitted. Table 7 shows experimental results using UCF-101 dataset, while Table 8 shows ones using LIRIS-HARL dataset. 

Table 7 shows that when the tIoU value was between 0.1 and 0.3, the model proposed in [40] showed higher performance; however, when tIoU > 0.5, whereby a higher accuracy was required for fine-level action detection, the performance of the proposed model was higher. Table 8 shows that our model outperforms the model proposed in [41] for all given ranges of tIoU. Furthermore, as tIoU increases, the difference in performance also increases.

Figure 8 illustrates the visualization of real results for the detection of actions in videos by using the proposed model. Figure 8a,b show the classification of two actions “BasketBallDunk” and “BasketBall” by using the pose information of the two actions occurring on a basketball court. Moreover, in Figure 8a, as the classification was performed properly until the time boundary that the action had occurred, high accuracy of action detection performance was confirmed. In Figure 8b, in the middle of the action detection result, the action was classified incorrectly as “FloorGymnastics”; this is an example for incorrect classification of an action because of confusion with the action in Figure 8c. In both cases (Figure 8b,c), the motions of raising both arms are shown. In the case of Figure 8c, the human detection was missed at the end of the video when the person turned around in the process of jumping. If the number of missed human detections increases, the spatio-temporal action-detection performance could decrease. In Figure 8d, the detected human pose information is not consistent with the real pose of a human in the image; this shows a good example of performing correct video-based action detection by using the surrounding background or object information, whereby the AC is performed using the visual features. 

## 6. Conclusions

In this paper, a deep neural network model was proposed for spatio-temporal action detection. To supplement the comprehensive coarse-level features obtained across a whole conventional image, fine-level features were used. Fine-level features contain feature information directly related with an individual performing an action. By using the pose information, i.e., fine-level features, the information of motions that compose an action could be obtained. Through experiments, this paper proved that when both coarse- and fine-level features are used simultaneously, a higher detection accuracy can be obtained than when using only coarse-level features. Furthermore, for the TRP process used in this study, a complementary two-stage region-proposal process was designed. The experiment proved that the accuracy and speed of action detection can be improved by filtering the regions not containing actions prior to generating anchor boxes. In the SRP process, humans were detected in a video by using the SSD, a conventional object-detection model. However, when a person assumed an abnormal pose or turned around, the detection was missed. To prevent missing detections, an object-detection model that can detect a human in various poses should be used, and for the case of missing detection, an effective tracking technique is needed to track the action. Moreover, for an action comprising various motions, the pose features can cause confusion, as confirmed through an experiment. To prevent this problem, more refined action definitions are needed for respective motion types. Through various experiments using the benchmark datasets, LIRIS-HARL and UCF-101, the proposed model demonstrated high performance and effectiveness.

## Figures and Tables

**Figure 1 sensors-19-01085-f001:**
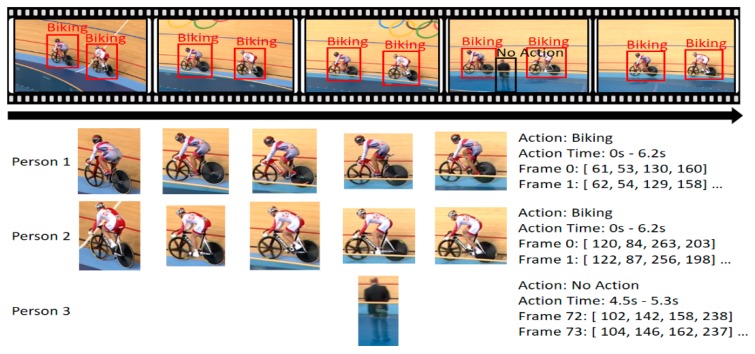
Examples of spatio-temporal action detection.

**Figure 2 sensors-19-01085-f002:**
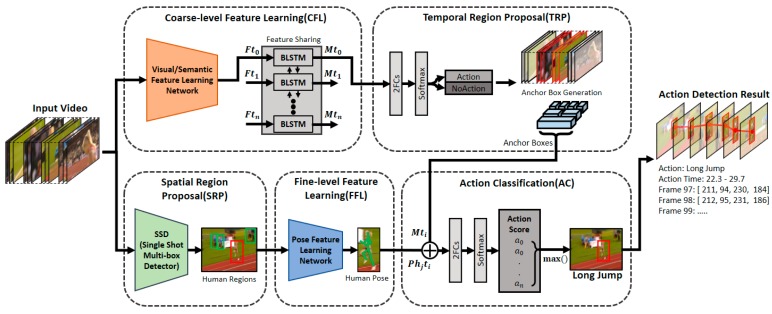
Spatio-temporal video-action-detection process.

**Figure 3 sensors-19-01085-f003:**
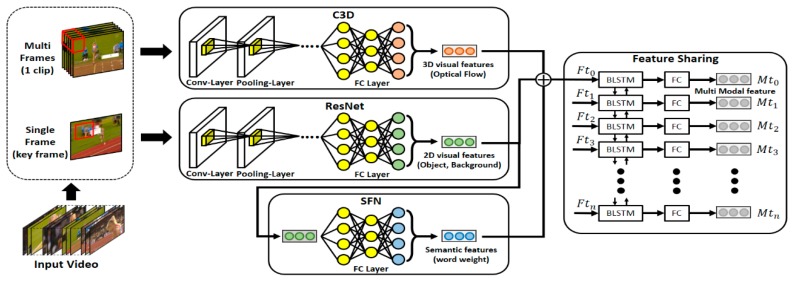
Coarse-level feature learning.

**Figure 4 sensors-19-01085-f004:**
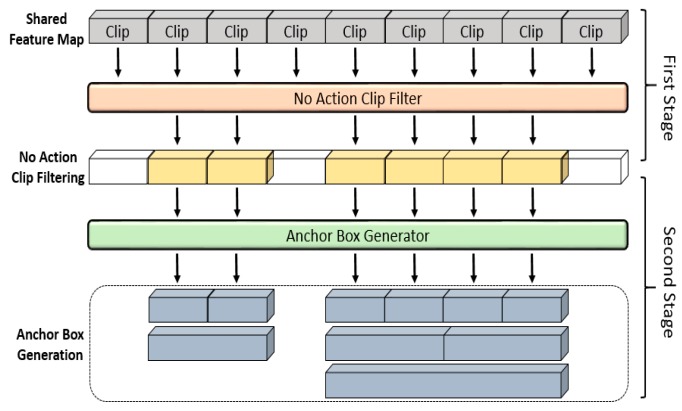
Temporal region proposal.

**Figure 5 sensors-19-01085-f005:**
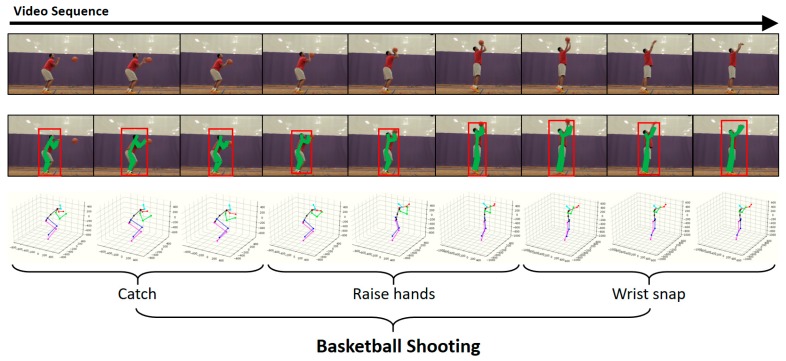
Pose estimation and action detection.

**Figure 6 sensors-19-01085-f006:**
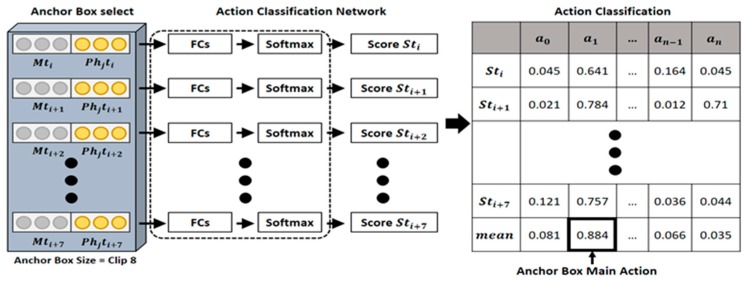
Action classification.

**Figure 7 sensors-19-01085-f007:**
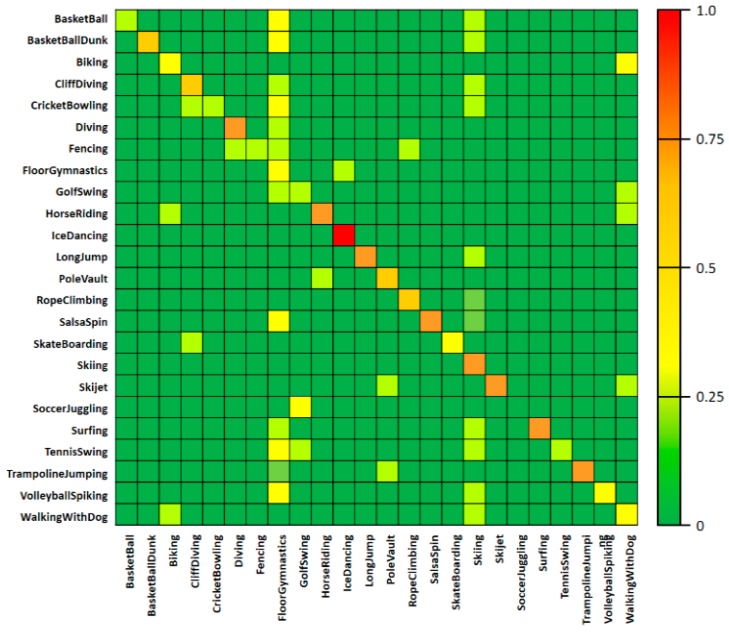
Confusion matrix for actions; UCF-101.

**Figure 8 sensors-19-01085-f008:**
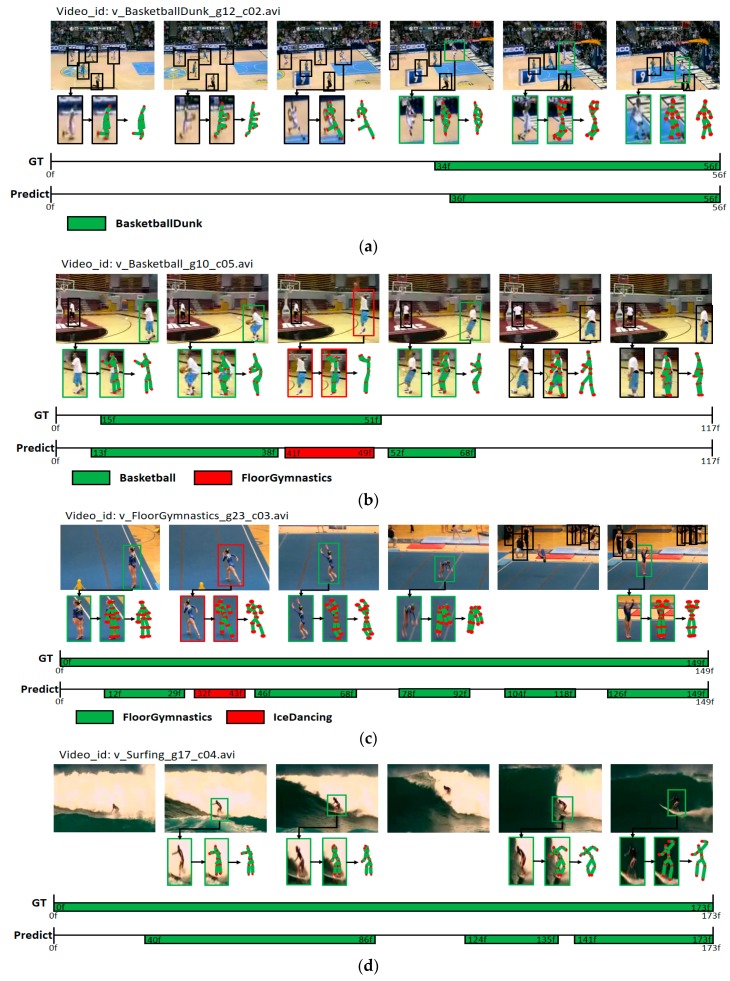
Visualization of action-detection results by using the spatio-temporal action-detection model; UCF-101. Figure (**a**–**d**) show results for videos each in UCF-101(24 classes). Ground truth activity segments (GT) are marked in green on the top line. Predicted activity segments (Predict) are marked in green for correct predictions and in red for wrong ones on the bottom line.

**Table 1 sensors-19-01085-t001:** Comparison of spatio-temporal action-detection performances (video-mAP) with the feature combinations on UCF-101 dataset.

	θ>0.1	θ>0.2	θ>0.3	θ>0.4	θ>0.5	mAP
CFL	60.9	53.1	42.0	27.8	11.9	39.1
FFL	42.8	30.6	21.1	11.3	5.4	22.2
CFL + FFL	62.3	56.7	51.5	44.8	36.6	50.3

**Table 2 sensors-19-01085-t002:** Comparison of spatio-temporal action-detection performances (video-mAP) with the feature combinations on LIRIS-HARL dataset.

	θ>0.1	θ>0.2	θ>0.3	θ>0.4	θ>0.5	mAP
CFL	40.1	31.3	23.4	15.9	11.4	24.2
FFL	53.3	47.8	36.1	27.8	20.2	37.0
CFL + FFL	59.4	55.5	42.9	35.4	27.6	44.1

**Table 3 sensors-19-01085-t003:** Comparison of spatio-temporal action-detection performances (video-mAP) with the feature sharing on UCF-101 dataset.

	θ>0.1	θ>0.2	θ>0.3	θ>0.4	θ>0.5	mAP
**MLP**	48.2	41.5	37.8	29.7	20.9	35.6
**LSTM**	60.5	51.1	46.6	38.4	29.6	45.2
**BLSTM**	62.3	56.7	51.5	44.8	36.6	50.3

**Table 4 sensors-19-01085-t004:** Spatio-temporal action-detection performance (video-mAP) depending on the increasing threshold of temporal region proposal scores (tTRPS) in the UCF-101 dataset.

		θ>0.1	θ>0.2	θ>0.3	θ>0.4	θ>0.5	mAP
**Single-Stage** **Temporal Proposal**	**tTRPS > 0.00**	55.7	49.1	44.2	35.4	30.2	42.9
**Two-sSage** **Temporal Proposal**	**tTRPS > 0.15**	58.3	53.2	48.4	39.7	33.8	46.6
	**tTRPS > 0.30**	62.3	56.7	51.5	44.8	36.6	50.3
	**tTRPS > 0.45**	60.5	55.6	50.2	41.3	34.5	48.4

**Table 5 sensors-19-01085-t005:** Action-detection time depending on the increasing tTRPS.

	tTRPS	Time(ms)
**Single-Stage** **Temporal Proposal**	tTRPS > 0.00	4085.0
**Two-Stage** **Temporal Proposal**	tTRPS > 0.15	67.2
	tTRPS > 0.30	56.3
	tTRPS > 0.45	50.4

**Table 6 sensors-19-01085-t006:** Comparison between action-detection performances of tIoU and UCF-101 dataset.

	**Basketball**	**Basketball Dunk**	**Biking**	**Cliff Diving**	**Cricket Bowling**	**Diving**	**Fencing**	**Floor Gymnastics**	**Golf Swing**	**Horse Riding**	**Ice Dancing**	**Long Jump**
θ>0.1	45.5	86.4	51.3	46.1	39.3	83.7	38.7	45.1	37.1	72.3	97.8	72.9
θ>0.2	38.6	86.4	45.9	48.7	36.4	76.7	25.8	41.9	31.4	61.7	97.8	64.8
θ>0.3	35.1	86.4	37.8	48.7	33.5	74.4	16.1	32.2	28.5	55.3	95.6	59.4
θ>0.4	30.5	81.0	27.0	46.1	32.9	62.7	12.9	29.0	25.7	46.8	89.1	51.3
θ>0.5	28.5	81.0	10.8	41.0	30.3	44.1	10.2	16.1	25.7	37.0	71.7	37.8
**Pole Vault**	**Rope Climbing**	**Salsa Spin**	**Skate Boarding**	**Skiing**	**Skijet**	**Soccer Juggling**	**Surfing**	**Tennis Swing**	**Trampoline Jumping**	**Volleyball Spiking**	**Walking WithDog**	**mAP**
82.0	60.0	100.0	42.8	78.9	82.1	12.8	93.5	22.7	76.6	81.8	55.5	62.3
64.1	52.0	100.0	39.2	76.3	71.4	10.2	87.0	22.7	63.3	78.7	47.2	56.7
53.5	52.0	100.0	32.1	60.5	57.1	7.6	74.1	22.7	60.0	78.7	38.8	51.5
40.5	44.0	83.3	20.7	47.3	50.0	5.1	74.1	20.4	56.6	66.6	36.1	44.8
30.4	32.5	80.1	18.5	35.7	25.0	2.5	70.9	18.1	46.6	63.3	21.1	36.6

**Table 7 sensors-19-01085-t007:** Comparison of action-detection performances between different models; UCF-101 dataset.

	θ>0.1	θ>0.2	θ>0.3	θ>0.4	θ>0.5
**Yu et al. [38]**	42.8	26.5	14.6	-	-
**Weinzaepfel et al. [39]**	51.7	46.8	37.8	-	-
**Peng et al. [40]**	77.3	72.8	65.7	-	30.8
**Saha et al. [41]**	76.1	66.3	54.9		
**Hou et al. [42]**	51.3	47.1	39.2		
**Our Model**	62.3	56.7	51.5	44.8	36.6

**Table 8 sensors-19-01085-t008:** Comparison of action-detection performances between different models; LIRIS-HARL dataset.

	θ>0.1	θ>0.2	θ>0.3	θ>0.4	θ>0.5
**Saha et al. [41]**	54.1	49.1	35.9	28.0	21.3
**Our Model**	59.4	55.5	42.9	35.4	27.6

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
