# Peer review of "Spatio-Temporal Action Detection in Untrimmed Videos by Using Multimodal Features and Region Proposals"

_sensors, 2019, doi:10.3390/s19051085_

Round 1
Reviewer 1 Report
# This paper tackles the full spatio-temporal action detection problem (line 11).
The paper is interesting and enjoyable to read, however it is not clear what the particular contribution of the authors is over the current state-of-the-art (see missing citations), and the experimental evaluation seems to be missing relevant datasets and evaluation metrics for action detection. For these reasons I do not think the paper should be considered for publication at this time. I encourage the authors to make amendments to the contributions section, and make clear what ideas are derived from previous works, and which ones they are bringing to the table. I would also encourage the authors to clearly specify the problem that they are tackling from the outset (single frame action detection, and temporal localization), so that the method and experiments can be appreciated in that context. Finally I encourage the authors to preform experiments that validate the parts of the method which are particularly novel and report those results in the experimental section.
# Missing citations:
- spatial and temporal detection of multiple co-occurring actions in a video:
"Deep Learning for Detecting Multiple Space-Time Action Tubes in Videos
Suman Saha, Gurkirt Singh, Michael Sapienza, Philip H. S. Torr, Fabio Cuzzolin"
- State of the art in per-frame action detection:
"Video Action Transformer Network
Rohit Girdhar, João Carreira, Carl Doersch, Andrew Zisserman"
- Using pose information as a feature for action classification:
PoTion: Pose MoTion Representation for Action Recognition
Vasileios Choutas, Philippe Weinzaepfel, Jérôme Revaud, Cordelia Schmid
line 240: "a person that performs an action cannot be known accurately in a video when many people appear"
line 253: "propose a method that tracks human region information"
See:
Incremental Tube Construction for Human Action Detection
Harkirat Singh Behl, Michael Sapienza, Gurkirt Singh, Suman Saha, Fabio Cuzzolin, Philip H. S. Torr
- Using space-time tube proposals:
Detect-and-Track: Efficient Pose Estimation in Videos
Rohit Girdhar, Georgia Gkioxari, Lorenzo Torresani, Manohar Paluri, Du Tran
# Experimental evaluation
- The datasets that are used for evaluation either contain one action category in each video, with multiple actors (UCF101), or a single action that extends the whole length of the video (JHMDB-21). As such there are more relevant datasets for evaluating action detection (LIRIS-HARL, AVA).
- For per-frame action classification performance, the state-of-the-art is benchmarked on larger and more challenging datasets such as AVA, it would be good to compare your method to the state-of-the-art on such a dataset.
- If the goal of this paper is to localize all multiple-action regions in a video (line 11), then why is 'frame-level' action detection relevant? Each frame-level detection is only a moment (part) of an action in a video. A single action is composed of multiple such moments (parts), and should be treated as such. Also, temporal detection is only part of the action detection problem, why not consider the 3D overlap between the action tubes in both space and time? Especially when considering multiple-action detection in each video, which is one of the claims of this paper.
- It would be good to see an ablation study to see the effects of different parts of the pipeline have on the overall action detection performance, in order to evaluate the effects of your particular contributions.
# Misc notes
- expand on the relationship between human pose an actions. Different types of actions can have the same human pose.
- method section is wordy, try to reduce the redundancy in text and write more concisely.
# typos:
line 21: "UCF 10"
Author Response
Please refer to the attached reply letter.

Reviewer 2 Report
The introduction can be improved. The authors list four contributions in the introduction. The fourth contribution can be merged with the first contribution since the goal of detecting human is for pose feature estimations as fine-grain features.
The current introduction naturally derives the first contribution, but for the second contribution of using BLSTM, it is not mentioned in the previous paragraphs and it is also not validated in the experiment sections for the comparison with single direction LSTM.
For the third contribution to generating proposals, the authors use the word of anchor which is used in regression-based proposal network to do coordinate regression. The authors might misuse this word. Also, these descriptions in section 3.2 (temporal region proposals), section 3.5 (Action classification) and Figure 6, don't describe clearly how the model realizes temporal localization. To my understanding, the localization is achieved by sliding window plus binary classification in the first stage. The second stage only uses "anchors" to average classification scores, and anchors don't take the role of localization. If I have any misunderstanding, please re-edit the text and clarify.
The experiments section can be re-organized to be more readable, with headings such as comparison with state-of-the-art, ablation studies etc.
The task is essentially spatio-temporal action detection, but the authors call it a different name "dense action detection". If I have misunderstood, please clarify.
Author Response

(The authors gave the same response as above.)

Reviewer 3 Report
The paper proposes a method for action detection and localization in long videos. Spatio-temporal region proposals are first learned and generated and then the detection and localization is performed based on the proposals. To distinguish different person targets in multiple detections, a fine-grained feature learning scheme is used to predict human poses. Extensive experiments are conducted on JHMDB21 and UCF-10 datasets, demonstrating the effectiveness of the proposed approach.
The reviewer has some minor comments on the manuscript:
(i) It is not clear that how the human poses are learned together with the network training since people usually do not have joint location information in action videos. How do the authors get the supervision to learn the human pose? Do they use a pre-trained model to first produce human-pose groundtruth? If so, the supervision is quite unreliable, as some of them would be wrong predictions. The authors need to clarify this more clearly.
(ii) The state of the art comparison only considers three approaches. However, there are many other approaches working on these two datasets. The authors should add more in the comparison to show the superior performance.
(iii) There is a limited review of related works. The authors should enrich the related work part. The reviewer only lists several of them here:
- Search Action Proposals via Spatial Actionness Estimation and Temporal Path Inference and Tracking, ACCV 2016
- Single Shot Temporal Action Detection, ACM MM 2018
- Recurrent tubelet proposal and recognition networks for action detection, ECCV 2018
Author Response

(The authors gave the same response as above.)

Round 2
Reviewer 1 Report
I feel the the authors did not sufficiently address the reviewer's comments. I would encourage the authors to spend more time to revise and update the manuscript before re-submitting.
Author Response
Comments of Reviewer #1
I feel the authors did not sufficiently address the reviewer's comments. I would encourage the authors to spend more time to revise and update the manuscript before re-submitting.
1. The paper is interesting and enjoyable to read, however it is not clear what the particular contribution of the authors is over the current state-of-the-art (see missing citations), and the experimental evaluation seems to be missing relevant datasets and evaluation metrics for action detection. I encourage the authors to make amendments to the contributions section, and make clear what ideas are derived from previous works, and which ones they are bringing to the table.
[Response]
We revised our contributions in the introduction section:
“1. In the proposed model, both coarse-level features and fine-level features are learned simultaneously for video action detection. Coarse-level features contain comprehensive information about the image, while fine-level features contain independent information for each person. ~”
2. I would also encourage the authors to clearly specify the problem that they are tackling from the outset (single frame action detection, and temporal localization), so that the method and experiments can be appreciated in that context. Finally I encourage the authors to preform experiments that validate the parts of the method which are particularly novel and report those results in the experimental section.
[Response]
The experiments have been reorganized and added to validate the parts of the proposed method:
Table 6 Comparison of action-detection performances (video-mAP) with the feature combinations
“~ In the first experiment, the action-detection performances between different combinations of features were compared to verify the effect of combining the coarse- and fine-level features. ~”
Table 7 Comparison of action-detection performances (video-mAP) with the feature sharing
“~ The second experiment was conducted to analyze the effect of the feature sharing process on the action-detection performance. ~”
Table 8&9 Action detection performance (video-mAP) depending on the increasing tTRPS
“~ The third experiment was conducted to analyze the effect of the proposed TRP process on the action-detection performance. ~”
3. Missing citations
- spatial and temporal detection of multiple co-occurring actions in a video:
"Deep Learning for Detecting Multiple Space-Time Action Tubes in Videos
Suman Saha, Gurkirt Singh, Michael Sapienza, Philip H. S. Torr, Fabio Cuzzolin"
- State of the art in per-frame action detection:
"Video Action Transformer Network
Rohit Girdhar, Joao Carreira, Carl Doersch, Andrew Zisserman"
- Using pose information as a feature for action classification:
PoTion: Pose MoTion Representation for Action Recognition
Vasileios Choutas, Philippe Weinzaepfel, Jerome Revaud, Cordelia Schmid
line 240: "a person that performs an action cannot be known accurately in a video when many people appear"
line 253: "propose a method that tracks human region information"
See:
Incremental Tube Construction for Human Action Detection
Harkirat Singh Behl, Michael Sapienza, Gurkirt Singh, Suman Saha, Fabio Cuzzolin, Philip H. S. Torr
- Using space-time tube proposals:
Detect-and-Track: Efficient Pose Estimation in Videos
Rohit Girdhar, Georgia Gkioxari, Lorenzo Torresani, Manohar Paluri, Du Tran
[Response]
The citations you had provided were added and the related work section was updated:
“~ Several works attempt to simultaneously localize not only temporal, but also spatial regions of action. The work in [40] tries to detect human regions appearing in each image of video and classify what type of actions are included in them. Recently, there are works [1,2] to localize the spatio-temporal area of an action by calculating associations between subsequent images.”
“~ On the other hand, learning effective features from video is also an important issue. Most of conventional works in video action detection use only coarse-level features extracted by CNN, which contain comprehensive information of an entire image. Recently, however, some studies attempt to make use of human pose information for detecting actions from a video. In the work of [26], the 2D positions of human joints are obtained in frame-by-frame, and then the trajectory information of individual joints is obtained by collecting joint positions over a video clip. However, 2D pose information is not enough to identify the exact type of actions contained in a video. In this paper, we use 3D pose information, which is more effective than 2D pose for video action detection. ~”
26. Vasileios, C.; Philippe, W.; Jerome, R.; et al. PoTion: Pose MoTion Representation for Action Recognition. IEEE Conf. Comput. Vis. Pattern Recognit. (CVPR), 2018.
40. Suman, S.; Gurkirt, S.; Michael, S.; et al. Deep Learning for Detecting Multiple Space-Time Action Tubes in Videos. British. Machine. Vis. Conf. (BMVC), 2016.
4. The datasets that are used for evaluation either contain one action category in each video, with multiple actors (UCF101), or a single action that extends the whole length of the video (JHMDB-21). As such there are more relevant datasets for evaluating action detection (LIRIS-HARL, AVA). For per-frame action classification performance, the state-of-the-art is benchmarked on larger and more challenging datasets such as AVA, it would be good to compare your method to the state-of-the-art on such a dataset.
[Response]
We, authors, also considered the AVA dataset to use for experiments. However, the AVA dataset does not include multiple actions in each video, so we decided that the AVA dataset is not suitable for our detection model. On the other hand, both the UCF-101 and the JHMDB-21 datasets have all properties required to show the power of our model. Moreover, two datasets were used commonly in the state-of-the-art works for video action detection. Therefore, we used only the UCF-101 and the JHMDB-21 datasets to evaluate the performance of the proposed model.
5. expand on the relationship between human pose an actions. Different types of actions can have the same human pose.
[Response]
We, authors, decided to use coarse-level features to identify multiple distinct actions with similar poses. Coarse-level features can provide additional information regarding surrounding background or related objects. We revised the fine-level feature learning section:
“~ However, using only pose information is not enough to identify multiple distinct actions with similar poses. Considering such cases, surrounding background or related objects must be considered together. Therefore, in this paper, the coarse-level and fine-level features are used together to obtain complementary effect. ~”

Round 3
Reviewer 1 Report
There are four main areas that need to be addressed for this work to be considered for publication, A, B, C and D. I encourage the authors to address each one of these areas.
Points to address before publication:
—-(A) Contributions.
1 Coarse and fine level features have been used extensively in the computer vision literature.
Is there something about the simultaneous learning of coarse and fine features which is novel?
Were there any necessary changes to the network architecture that ensured good performance?
3 Have you experimentally validated how the two-stage temporal region proposal method improves detection accuracy?
--(B) Missing relevant datasets for evaluation.
The datasets that are used for evaluation either contain one action category in each video, with multiple actors (UCF101), or a single action that extends the whole length of the video (JHMDB-21), temporal detection is redundant.
No dataset used in this paper contains multiple actions and multiple people in untrimmed videos.
As such there are more relevant datasets for evaluating action detection (LIRIS-HARL, AVA).
—-(C) Evaluation metrics must match the stated problem
The paper title and introduction clearly state the problem on spatio-temporal action detection. If the task you consider in this paper is "spatio-temporal action detection", then the appropriate evaluation metric to use would be the 3D overlap between the action tubes in both space and time.
It seems as if this paper tackles the task of "single-frame action detection in video" (evaluated with single-frame mAP) and "temporal localization" (evaluated with IoU of temporal regions) separately. If this is the case, then it needs to be clearly stated in the title and introduction, and the pros/cons and limitations need to be discussed.
Note that temporal localization is useless in JHMDB-21 as the videos are trimmed, so additional dataset/experiments are needed to validate temporal localization (see (B)).
—-(D) Clarification of evaluation metrics
In table 11, the comparison to the state-of-the-art, the metric used is “tIOU”.
This does not seem to be the same metric used in Gang etal [37] (eq 6). Can you clarify this?
Gang etal [37] report results for theta > 0.1. Where did you obtain the results for other thresholds? 0.2 and 0.3?
In Philippe etal, the define: " The IoU between two tracks is defined
as the IoU over the temporal domain, multiplied by the average of the IoU between boxes averaged over all overlapping frames. Duplicate detections are considered as “incorrect”."
Is this the metric you are using?
Peng etal [39] report video-mAP of 50.39 42.27 32.70 respectively; where did you get the values you report in the paper?
With regards to Suman etal, the results reported are: 76.57 66.75 55.46 46.35 35.86 26.79. Could you clarify where you obtained the numbers you report in the paper from?
Can you report and compare results using the same metrics on the came thresholds up to theta > 0.6?
More recent results to compare to can be seen here:
I encourage the authors to get the same evaluation metrics and scripts for a fair comparison.
So far the reported results do not show an improvement compared to other state-of-the-art methods.
Adding the latest papers, and clarifying the numbers and metrics will help the reader to understand the benefits of the various works.
Author Response
Please refer to the uploaded reply file.
